# Functional specialization within the inferior parietal lobes across cognitive domains

Ole Numssen[1], Danilo Bzdok[2,3][†]*, Gesa Hartwigsen[1][†]*

[1]Lise Meitner Research Group Cognition and Plasticity, Max Planck Institute for Human Cognitive and Brain Sciences Leipzig, Leipzig, Germany; [2]Department of Biomedical Engineering, McConnell Brain Imaging Centre, Montreal Neurological Institute, Faculty of Medicine, McGill University, Montreal, Canada; [3]Mila - Quebec Artificial Intelligence Institute, Montreal, Canada

**Abstract** The inferior parietal lobe (IPL) is a key neural substrate underlying diverse mental processes, from basic attention to language and social cognition, that define human interactions. Its putative domain-global role appears to tie into poorly understood differences between cognitive domains in both hemispheres. Across attentional, semantic, and social cognitive tasks, our study explored functional specialization within the IPL. The task specificity of IPL subregion activity was substantiated by distinct predictive signatures identified by multivariate pattern-learning algorithms. Moreover, the left and right IPL exerted domain-specific modulation of effective connectivity among their subregions. Task-evoked functional interactions of the anterior and posterior IPL subregions involved recruitment of distributed cortical partners. While anterior IPL subregions were engaged in strongly lateralized coupling links, both posterior subregions showed more symmetric coupling patterns across hemispheres. Our collective results shed light on how under-appreciated hemispheric specialization in the IPL supports some of the most distinctive human mental capacities.

*For correspondence:
danilo.bzdok@mcgill.ca (DB);
hartwigsen@cbs.mpg.de (GH)

[†]These authors contributed equally to this work

Competing interests: The authors declare that no competing interests exist.

## Introduction

Many cognitive processes are realized by spatially distributed neural networks in the human brain. The inferior parietal lobe (IPL) is a heteromodal convergence zone of various brain networks that is central to realizing key cognitive operations across different levels of the neural processing hierarchy (*Kernbach et al., 2018*; *Seghier, 2013*). These mental operations include lower level processes, such as spatial attention, as well as higher level processes that are distinctly elaborate in the human species, like semantic memory and modes of social exchange.

During primate evolution, the IPL has probably undergone remarkable expansion and functional reorganization (*Van Essen and Dierker, 2007*; *Xu et al., 2020*). The emergence of new areas inside the human IPL has been seriously considered (*Mars et al., 2013*), while its precise correspondence to nonhuman homologues is still debated (*Seghier, 2013*). The evolutionary trajectory of the IPL, as part of the recent cortical expansion in humans may bear important relationships to language ability, future planning, problem solving, and other complex mental operations (*Dohmatob et al., 2020*; *Seghier, 2013*) at which humans excel.

Diverse cognitive capacities have been linked to neural activity in the IPL. Among these, spatial attention is crucial in an ever-changing environment that requires rapid behavioral adaptation (*Corbetta et al., 2008*). Systematic reviews have identified the right IPL as a key region for visuospatial attention (*Corbetta and Shulman, 2002*). Accordingly, neurological damage of the right IPL entails a clinical condition called hemi-neglect: the failure to orient visual attention to the

contralesional side (*Morrow and Ratcliff, 1998*). In contrast to visuospatial attention, language function is widely accepted to lateralize to the (dominant) left hemisphere (*Friederici, 2017*). Tissue damage to the left temporo-parietal cortex has been reported, for decades, to cause impairments of semantic processing – key to reading and other elaborate forms of language comprehension (*Binder et al., 2009*; *Hartwigsen et al., 2016*; *Seghier, 2013*). Semantic processing is instrumental for the human ability to contextualize and act according to the meaning of objects, events, and situations (*Lambon Ralph and Patterson, 2008*).

Moreover, semantic concepts are an integral component of numerous high-level cognitive operations, and are suspected to contribute to both social cognition and language from early development to adulthood (*Binder and Desai, 2011*). Semantic processing is presumably closely intertwined with social cognition. This view receives support from previous reports on spatial overlap between both cognitive domains in the left IPL (*Bzdok et al., 2016*; *Mars et al., 2011*; *Seghier, 2013*). However, many human neuroimaging experiments have shown that advanced social cognitive functions, such as the capacity to infer others' thoughts, beliefs, and behavioral dispositions, tend to engage the IPL in both hemispheres (*Bzdok et al., 2012*; *Bzdok et al., 2016*).

Substantiating these earlier hints at functional specialization of left versus right IPL, both regions also differ on the structural level. Architectural heterogeneity is indicated by cytoarchitectonic borders and gyral differentiation (*Caspers et al., 2008*) as well as anatomical fiber bundle connections as quantified by fiber tractography (*Caspers et al., 2011*). Such heterogeneity in the IPL is not only found between individuals, but also between the left and right hemisphere (*Toga and Thompson, 2003*). This structural scaffold has the potential to support the unique neurocognitive properties of the left and right IPL.

Finally, the IPL harbors a major hub of the transmodal association network (*Braga et al., 2019*) ('default mode network') and is densely connected with other cortical key areas for various functions. This is reflected in intimate functional interactions with several other large-scale brain networks as measured by intrinsic functional connectivity (*Kernbach et al., 2018*; *Buckner and DiNicola, 2019*). Despite indications in favor of lateralized functional specialization for human-defining cognitive domains, it remains unclear how the left and right IPL interact with distributed neural networks to realize advanced cognitive operations like social and semantic processes.

For these reasons, our study capitalized on cognitive neuroimaging experiments that tap on multiple functional domains. The combination of an arsenal of data-driven analysis techniques allowed zooming in on the hemisphere-specific functional specialization and brain-wide interaction profiles of the IPL and its segregated components. Our well-matched experimental paradigms prompted attentional reorienting, lexical decisions, and mental perspective taking – archetypical processes that exemplify the broader cognitive domains attention, semantics, and social cognition. The direct task comparison in the same sample of subjects enabled us to identify functional specialization and involvement of common mechanisms across attentional, semantic, and social realms of human cognition. Our unique characterization of IPL function, subspecialization, and interaction patterns provides key insights into cognitive operations that underlie elaborate forms of human interaction and communication.

## Results

### IPL recruitment in different cognitive tasks: neural activity responses

We designed three different tasks to probe the three domains of interest. An attentional reorienting task required subjects to make spatially congruent button presses in response to visually presented cues. A lexical decision task probed visual word-pseudoword decisions and a mental perspective taking task prompted decisions on the mental beliefs of others (*Figure 1—figure supplement 1* for the experimental design and behavioral results).

We first examined neural activity responses for each task separately by contrasting each target condition (attentional reorienting: *invalid,* lexical decision task: *word*, perspective taking: *false belief*) with its respective control condition (*valid*, *pseudoword*, *true belief*). In this analysis, all tasks showed increased neural activity in a widespread set of brain regions, including the IPL.

### Attentional reorienting task

Relative to the control condition, we found that the attentional reorienting condition significantly modulated neural activity in several nodes of the default mode network, including the right inferior parietal lobes (IPL) (supramarginal gyrus) and to a lesser extent also the left IPL (supramarginal gyrus), as well as the bilateral precuneus. Key nodes of the dorsal attention network were also activated, including the left frontal eye field and superior parietal lobe.

### Lexical decision task

Compared to pseudoword processing, real-word processing predominantly activated left-hemispheric parts of the default mode network, including the IPL (angular gyrus) and middle cingulate gyrus, as well as the superior frontal gyrus.

### Social cognition task

As expected, the results for the social cognition task were less strong (*Rothmayr et al., 2011*; *Sommer et al., 2010*). Peak activations for false belief relative to true belief trials (p < 0.001 uncorrected) encompassed bilateral areas, including the supplementary motor cortex, inferior frontal gyrus and precentral gyrus. To a lesser extent, nodes of the default mode network were also engaged, including the left precentral gyrus, right IPL (angular gyrus), and bilateral precuneus. For details on task-specific activity patterns, see *Figure 1* and *Supplementary file 1*.

To summarize neural activity responses in the IPL, we found strong upregulation of the right supramarginal gyrus (x, y, z = 57,–48, 24; T = 9.68) during attentional reorienting, left angular gyrus (x, y, z = −51,–72, 28; T = 8.00) during lexical decisions, and right angular gyrus (51, -60, 31; T = 5.15) during perspective taking. Standard mass-univariate analyses showed that all three tasks recruited areas within the larger IPL region in at least one hemisphere.

## Functional parcellation of the IPL in two subregions

To isolate coherent structure-function mappings from neural task responses, we used data-driven clustering algorithms to separate the cytoarchitectonically constrained region of interest (ROI) in the IPL into subregions (see Materials and methods for details). According to the applied cluster validity criteria, a two-cluster solution was indicated to be optimal in each hemisphere considering different cluster numbers. The choice of two final clusters was based on majority votes yielding 97.1% and 78.8% agreement across 25 distinct cluster validity criteria (cf. Materials and methods) for left and right IPL, respectively (*Table 1*). The three-cluster solution emerged as the second best choice (left IPL: 2.8% agreement, right IPL: 21.2% agreement).

To investigate the possible influence of general motor responses within the IPL on our results, we estimated a third set of GLMs ($GLM_{cond+RT}$), which was based on the initial $GLM_{cond}$ with one additional regressor that modeled reaction times for all three tasks. This reaction time regressor captured unspecific motor preparation and execution components that were not of interest here. The task-dependent BOLD responses closely resembled the results from the original analysis (*Figure 2—figure supplement 2*). The data-driven clustering based on these results selected a similar parcellation solution with comparable certainty. 95.3% and 73.0% of the majority votes selected a two cluster solution for the left and right IPL respectively, based on $GLM_{cond+RT}$. 82 voxels in the left IPL and 73 in the right IPL were unstable across random initializations. The cluster similarity across solutions is illustrated in *Figure 2—figure supplement 3*. Due to the high similarity of both solutions, all following analyses were performed based on the parcellation results from the more parsimonious model $GLM_{cond}$.

For all later analysis steps, 74 (out of 1102) voxels in the left and 84 (out of 1123) in the right IPL ROI were removed from the final solution because their cluster assignments were unstable across the random centroid initializations of the k-means algorithm. All discarded voxels were located at the cluster borders. The IPL parcellation procedure led to one anterior subregion (left hemisphere: L-ant, right hemisphere: R-ant) and one posterior subregion (L-post, R-post) in each hemisphere (*Figure 2B*).

Hemispheric asymmetries in cluster assignment were evaluated by quantifying topographical overlap after flipping the right-left axis of the four derived IPL subregions (*Figure 2—figure supplement 2*). Around two thirds of the voxels composing the IPL ROI (left: 64.98%, right: 66.31%) were

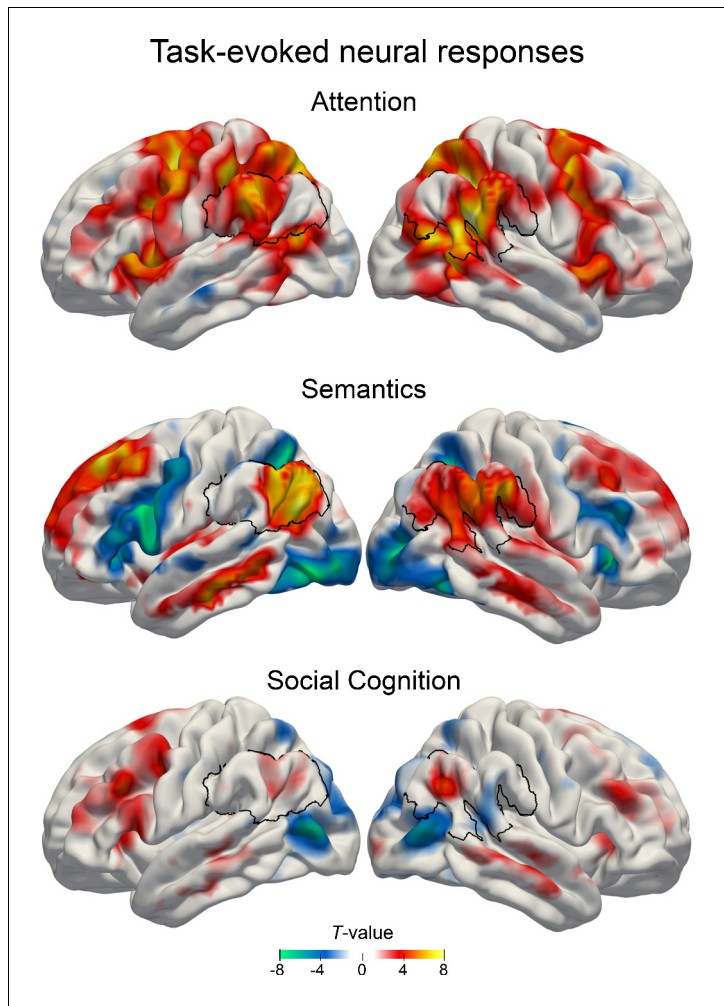

**Figure 1.** Main effects of estimated neural activity in the three cognitive domains. Task-dependent BOLD responses were quantified first using common mass-univariate analyses at the whole-brain level. Attentional reorienting task: *invalid > valid* contrast. Lexical decision task: *word > pseudoword* contrast. Perspective taking task: *false belief > true belief* contrast. Colors indicate unthresholded T-values for display purposes. Warm colors: higher GLM beta estimates for the target conditions. Cold colors: higher GLM beta estimates for the control condition. Black outline: region of interest in the inferior parietal lobe.

The online version of this article includes the following figure supplement(s) for figure 1:

**Figure supplement 1.** Experimental design and behavioral results.

**Table 1.** Task-derived IPL subregions.

| Hemisphere | Position | Label | Volume | Center of mass | Cytoarchitectonic area |
|---|---|---|---|---|---|
| Left | Anterior | L-ant | 11.6 cm$^3$ | −57,−32, 30 | PF, PFop, PGcm, PFt |
| Left | Posterior | L-post | 20.7 cm$^3$ | −48,−61, 33 | PGa, PGp, PFm, PF |
| Right | Anterior | R-ant | 12.9 cm$^3$ | 58,−29, 28 | PF, PFop, PGcm, PFt |
| Right | Posterior | R-post | 19.8 cm$^3$ | 51,−57, 32 | PGa, PGp, PFm |

*Note:* Cytoarchitectonic assignment was performed with the SPM Anatomy Toolbox. Center of mass is given in MNI coordinates (x, y, z).

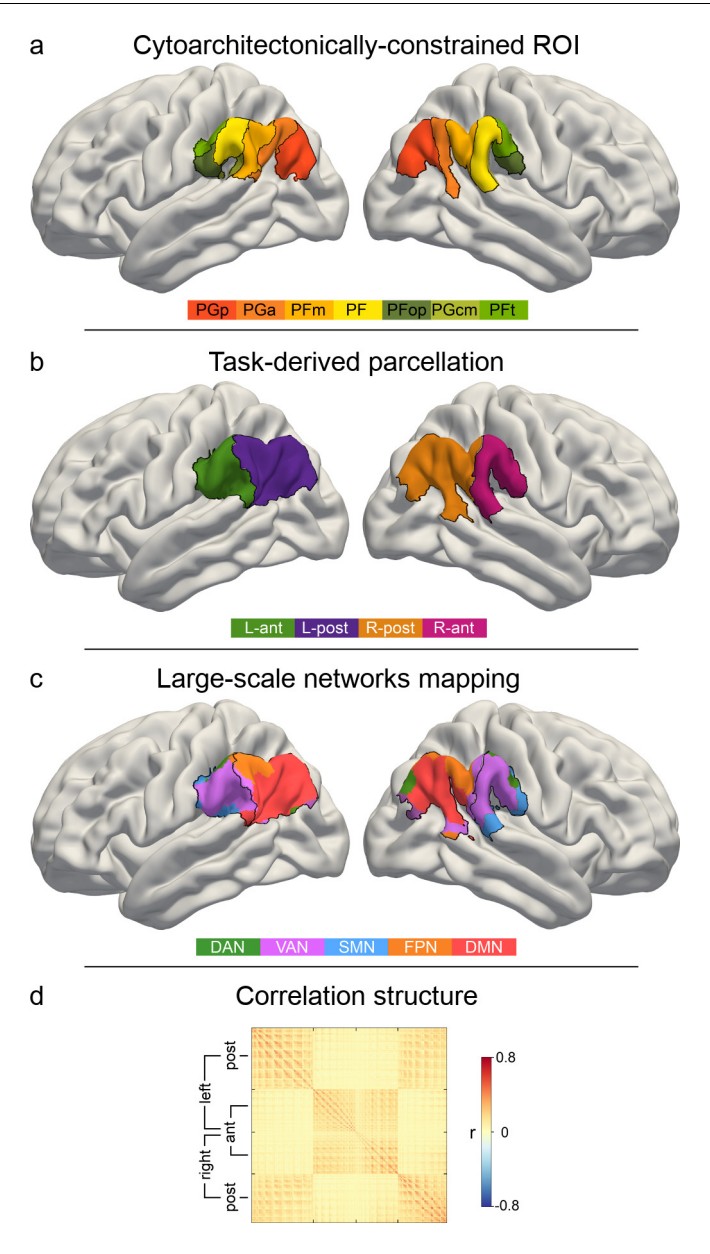

**Figure 2.** IPL parcellation extracted from neural activity responses across attention, semantics, and social cognition. Neural activity estimates from all three task experiments were pooled to achieve a data-driven segregation of the inferior parietal lobe, separately in each hemisphere. (**a**) Cytoarchitectonic boundaries defined the contours of our region of interest (ROI) (*Zilles and Amunts, 2010*). (**b**) The derived ROI was submitted to automatic parcellation into subregions to capture neural activity profiles of the three domains in each hemisphere. L-ant: left anterior subregion. L-post: left posterior subregion. R-ant: right anterior subregion. R-post: right posterior subregion. (**c**) Mapping of large-scale brain networks (*Yeo et al., 2011*) to the ROI. Posterior subregions are predominantly populated by the default mode network (DMN), while the anterior regions mainly host the ventral attention network (VAN). DAN: dorsal attention network. FPN: fronto-parietal network. SMN: somatomotor network. (**d**) Similarity matrix of voxel-wise neural activity estimates of single trials, reordered according to (**c**). The similarity structure reproduces the parcellation results.

The online version of this article includes the following figure supplement(s) for figure 2:

**Figure supplement 1.** The final subregion solution shows a high degree of symmetry between hemispheres.
**Figure supplement 2.** Task-evoked neural responses in the IPL region with explicit modeling of motor responses.
**Figure supplement 3.** Explicitly modeling motor responses yields a similar clustering solution.

spatially congruent across hemispheres. Considering these corresponding voxels contained in both left and right IPL ROI, subregion-specific across-hemisphere congruency was nearly perfect (L-ant: 88.97%; L-post: 100%; R-ant: 100%; R-post: 93.48%). The data-driven IPL parcellation into two subregions in each hemisphere served as the basis for all subsequent analyses.

## Task activity responses are predictive for different cognitive domains

Building on the obtained functional parcellation of the IPL into two core subregions that show different activation patterns for the probed cognitive domains, we investigated task-dependent specialization of IPL function using tools from machine learning. By deploying a linear predictive pattern-learning algorithm, we solved the classification problem of assigning task-membership from single experimental trials based on the subregion-level neural activity changes. We evaluated the quality of the predictive model via leave-one-subject-out cross-validation, that is, by testing the fitted model on data from a subject that was not used during model building. Our results indicate that neural activity response patterns from our IPL subregions alone carried relevant information that was granular enough to allow for successful task classification significantly above chance. The logistic predictive model was cross-validated in a one-versus-rest scheme to attempt classifying trial brain scans from hold-out subjects. This machine learning pipeline resulted in an overall out-of-sample classification accuracy of 52.11% (chance: 33.33%). Note that this cross-validation scheme leads to conservative accuracy estimates, as the classifier is tested on brain activity information from subjects it has not seen before. Therefore, this approach provides a reliable estimate for the expected generalization in neuroimaging subjects scanned in the future (*Bzdok and Yeo, 2017*). Prediction accuracy reached 31.09% for the attentional reorienting task, 59.24% for the lexical decision task and 66.41% for the perspective taking task.

Relative to the other two experimental tasks, increased neural activity in the right anterior subregion and less neural activity in the left anterior subregion was associated with the attentional reorienting task when considering multivariate patterns distributed across all IPL subregions (see *Figure 3*). In contrast, successful trial classification pertaining to lexical decisions was driven by neural activity increases in the left anterior subregion and neural activity decreases in the right anterior subregion, relative to the other two tasks. Finally, neural activity within both posterior subregions carried salient information that was instrumental in successfully discriminating trial brain scans recorded during perspective taking.

## Task-specific network connectivity resembles task complexity

We subsequently delineated the task-dependent profiles of how IPL subregions are functionally coupled with regions outside the IPL ROI. For this purpose, we assessed functional connectivity between the four IPL subregions and other cortical brain regions. By means of non-parametric permutation testing, we tested for statistically significant coupling differences (*Figure 4* and *Figure 4—figure supplement 1*). The connectivity profiles between the four IPL subregions and large-scale brain networks were significantly different between tasks. The anterior subregions showed strong degrees of hemispheric lateralization and domain specificity. In contrast, the posterior subregions showed less lateralization and a higher degree of across-network coupling.

### Task-specific connectivity profiles for anterior IPL subregions

During attentional reorienting, the right anterior IPL subregion was stronger involved in cortex-wide connectivity than its left-hemispheric counterpart (*Figure 5A*, left column; *Figure 5D*). The visual network and regions belonging to the dorsal attention network emerged as its preferred coupling partners (*Figure 4—figure supplement 1*). The left anterior subregion was less engaged in cortical connectivity, and showed the strongest coupling with the dorsal attention network and the fronto-parietal control network. Compared to the other two functional domains, lexical decisions led to less cortex-wide connectivity, most strongly with left hemispheric dorsal attention network regions. In contrast, the perspective taking connectivity profiles for the left and right anterior subregions showed coupling with various networks. These IPL subregions revealed the strongest coupling patterns with left-hemispheric default mode network and bilateral fronto-parietal control network regions.

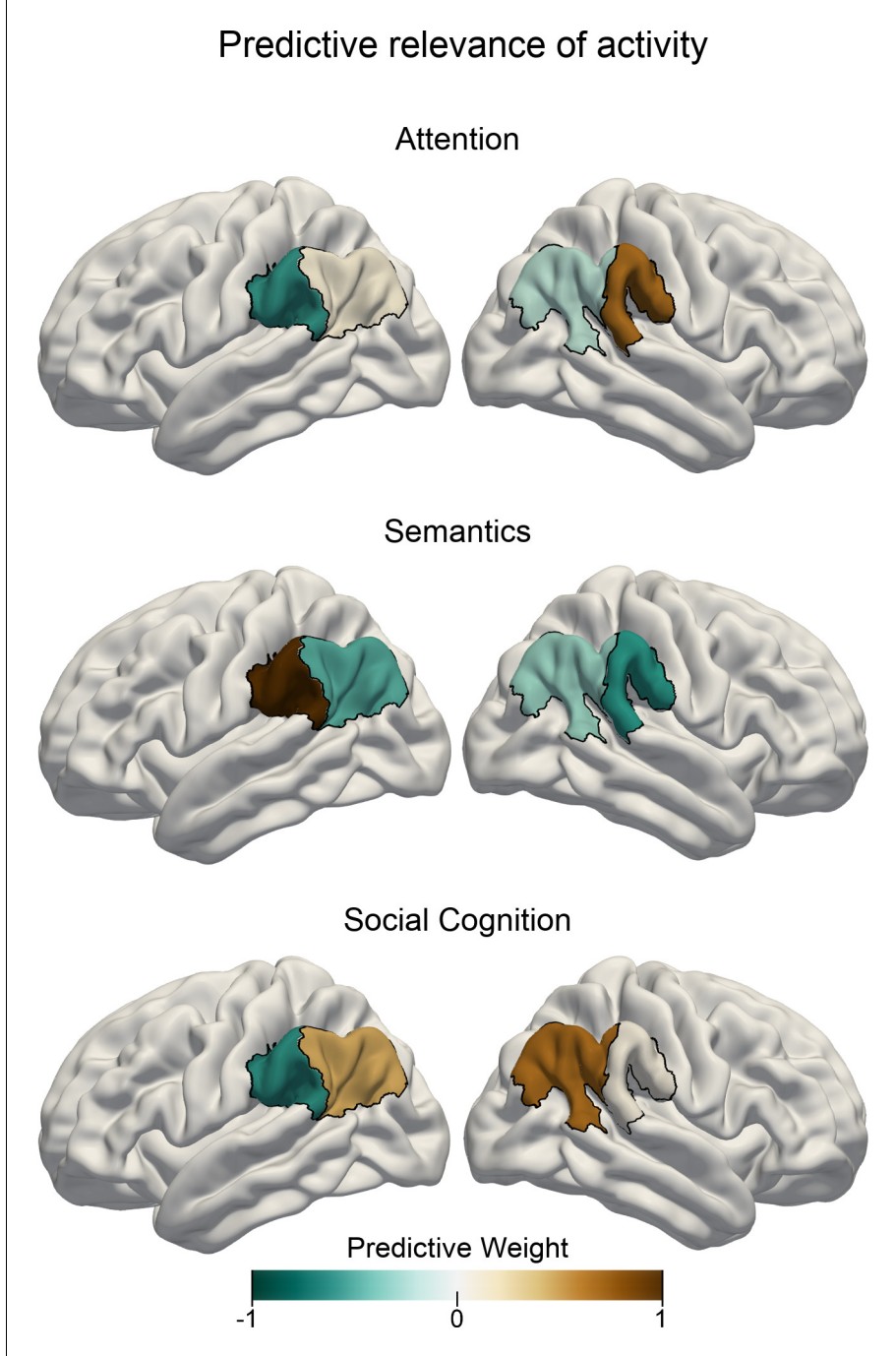

**Figure 3.** Task-specific predictive contributions of IPL subregions. Pattern-learning algorithms extracted predictive rules from neural activity estimates aggregated in the left vs. right anterior vs. posterior IPL subregions from the three target experimental conditions for attentional reorienting (top), lexical decisions (center), and perspective taking of others' mental states (bottom). Colors show the predictive signature with relative contributions of each of the four IPL subregions in detecting the presence of the three cognitive states from neural activity responses. A more positive subregion weight (brown color) for a given task implies that neural activity from this subregion carried information that increased the probability of a specific task being represented in trial brain scans. A leave-one-subject-out cross-validation was implemented to fit the predictive model. Negative values (cold colors) denote driving the prediction decision toward the respective other tasks.

The online version of this article includes the following figure supplement(s) for figure 3:

**Figure supplement 1.** Neural activity estimates for the target and control conditions of the three tasks.

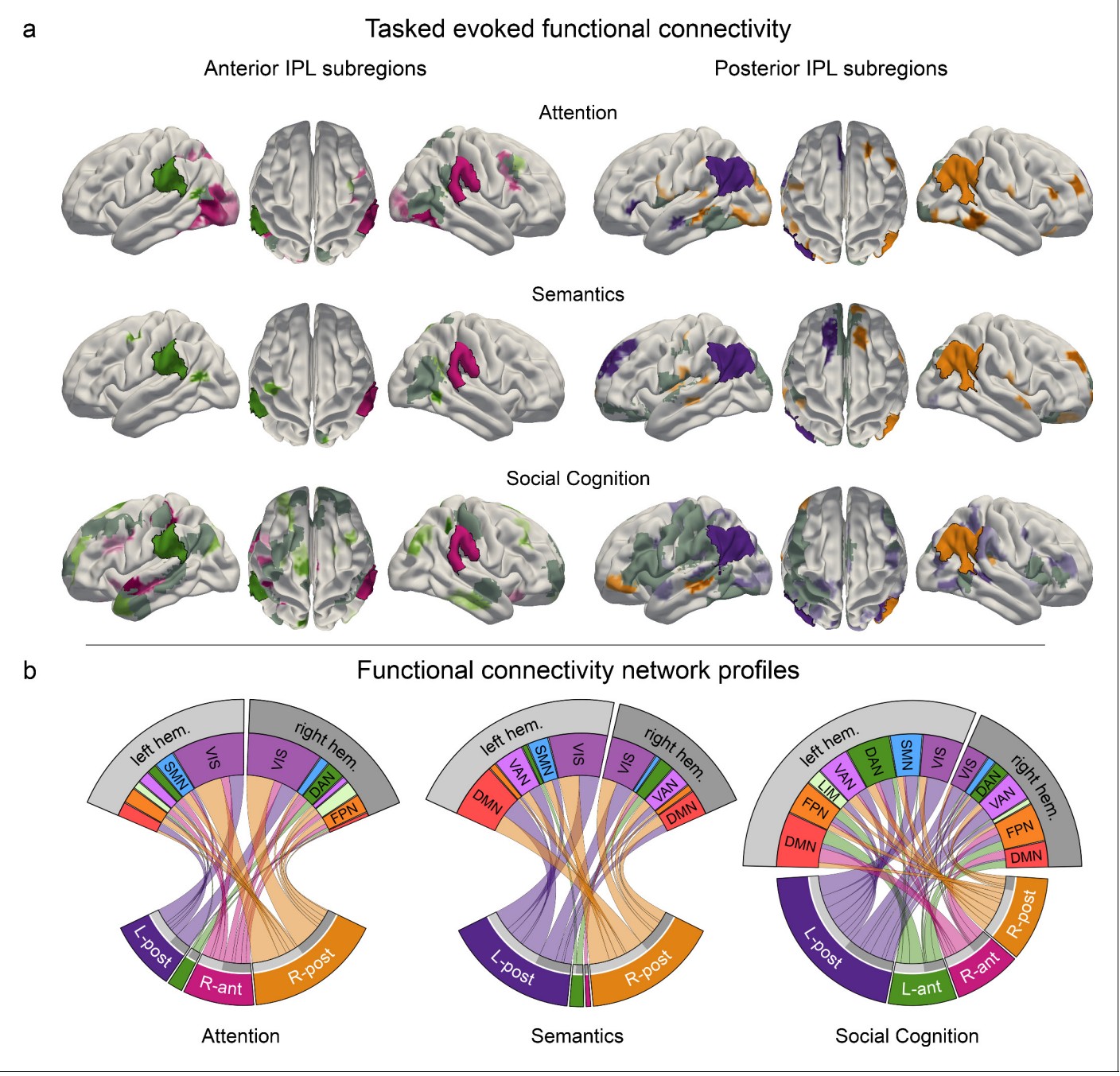

**Figure 4.** Task-induced shifts in cortex-wide functional connectivity. Task-dependent functional connectivity profiles of the four IPL subregions with other coupling partners, compared to the respective other tasks. (a) Task-specific correlation between anterior (left) and posterior (right) IPL subregions and brain-wide cortical regions. All results are statistically significant at $p < 0.05$, tested against the null hypothesis of indistinguishable connectivity strength between a given IPL subregion and the rest of the brain across all three tasks. L-ant: left anterior IPL subregion; L-post: left posterior IPL subregion; R-ant: right anterior IPL subregion; R-post: right posterior IPL subregion. Dark gray: mutual connectivity target. (b) Task-specific connectivity profiles of the four IPL subregions with seven large-scale brain networks (*Yeo et al., 2011*). DAN: dorsal attention network. DMN: default mode network. FPN: fronto-parietal network. LIM: limbic network. VAN: ventral attention network. SMN: somatomotor network. VIS: visual network. The online version of this article includes the following figure supplement(s) for figure 4:

**Figure supplement 1.** Subregion-specific functional connectivity.

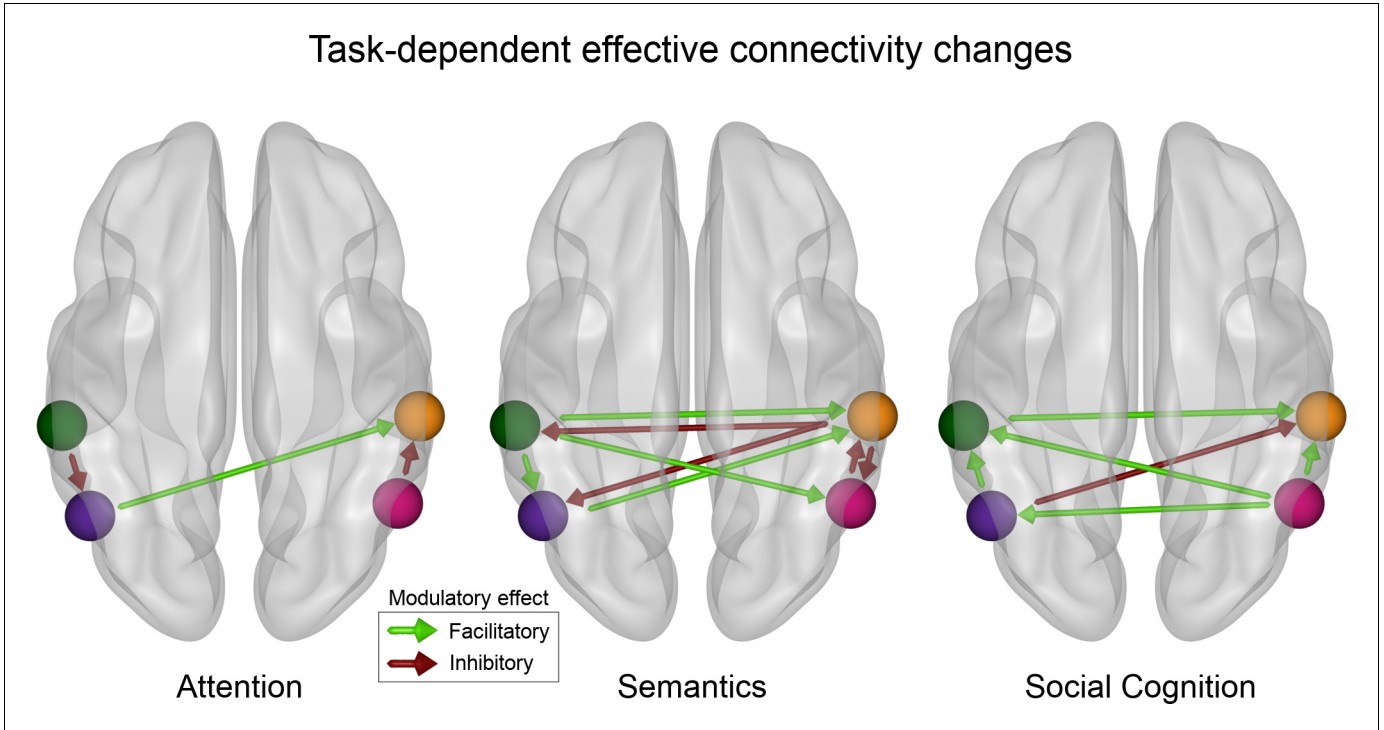

**Figure 5.** Tasks cause different effective connectivity modulations among IPL subregions. The modulation assessed by dynamic causal modeling (DCM) of directed connectivity among the four IPL subregions differs between attentional reorienting, lexical decisions, and perspective taking. Only significant modulatory parameters ($\alpha \leq 0.01$) are shown (based on the non-parametric permutation analysis). Attentional reorienting induces comparatively simple connectivity modulations, while both other tasks are characterized by richer changes in the modulatory influences between subregions. Lexical decisions increase facilitatory influences from the left to the right hemisphere and inhibition from the right to the left. Perspective taking causes bilateral increases in the facilitatory influence from posterior to anterior subregions. Green: DCM node in the left anterior subregion (L-ant). Violet: DCM node in the left posterior subregion (L-post). Orange: DCM node in the right anterior subregion (R-ant). Pink: DCM node in the right posterior (R-post) subregion.

### Task-specific connectivity profiles for posterior IPL subregions

During attentional reorienting, specific cortical connectivity shifts emerged for the right posterior subregion rather than the left posterior subregion of the IPL ROI (*Figure 5A*, right column; *Figure 5D*). Relative to the other two tasks, attentional reorienting led to significantly enhanced functional coupling with the visual network and left-hemispheric parts of the somatomotor network, as well as increased connectivity with right-hemispheric parts of the fronto-parietal control network (*Figure 4—figure supplement 1*). Processing lexical decisions in turn was specifically characterized by coherent connectivity increases between the posterior IPL subregions and the bilateral visual network, the default mode network and the ventral attention network. Finally, engagement in perspective taking engendered significantly stronger functional connectivity for the left posterior subregion and extended parts of several large-scale brain networks. While the left posterior subregion showed similar connectivity strength to bilateral parts of large-scale networks, the right posterior subregion mainly interacted with contralateral parts. The posterior IPL regions showed connectivity links with multiple coupling partners, including regions belonging to the visual network, to the ventral and dorsal attention network, and to the somatomotor network.

### Causal interactions between IPL subregions are task-dependent

To complement our findings on the task-evoked shifts in broader cortical connectivity profiles, we used dynamic causal modeling (DCM) to identify directed task-specific connectivity modulations between IPL subregions. The DCM optimization procedure provided one Bayesian parameter average group-level model. All modulatory parameters of this optimal model exceeded zero given a 95% confidence interval. Permutation-based tests provided evidence for task-specific differences in

the emerging modulatory configurations among the IPL subregions (*Figure 5*, see *Table 2* for modulatory parameter estimates between subregions and *Supplementary file 2* for other model parameters).

Explicitly probing modulatory effects on directed connectivity within the IPL, attentional reorienting was characterized by a simple coupling configuration, with three significant modulations: An increased inhibitory influence from the left anterior to the left posterior subregion, a facilitatory drive from the left posterior to the right anterior subregion, and an inhibitory influence from the right posterior to the right anterior subregion (*Figure 5*, left). The attention coupling topology was distinctively closer to the intrinsic connectivity than effective modulation patterns underlying both other domains. Both other tasks prompted more complex patterns of coupling modulations among IPL subregions. During lexical decisions, we found strong interhemispheric modulations, both facilitatory and inhibitory, between IPL subregions. Strikingly, task-related connectivity increases from the left anterior subregion to all other subregions were facilitatory. In contrast, the right anterior subregion had an inhibitory influence on all other subregions (*Figure 5*, center). Perspective taking was mainly characterized by increased facilitatory influences between IPL subregions, with the right posterior subregion exerting a facilitatory influence on all other subregions (*Figure 5*, right panel). Interhemispheric interactions between subregions were also mainly facilitatory during perspective taking (*Figure 5*, left).

To summarize, the three cognitive domains of interest were characterized by fundamentally different motifs of effective connectivity modulations among IPL subregions. A core observation relies on a simple directed connectivity profile for attentional reorienting and more complex directed connectivity interactions between IPL subregions for both higher-level tasks.

**Table 2.** Task-specific effective connectivity modulation.

| Source subregion | | Target subregion | Modulation strength | p-value |
|---|---|---|---|---|
| Attention | | | | |
| L-ant | → | R-post | −1.93 | 0.0017 |
| L-post | → | R-ant | 3.01 | 0.0033 |
| R-post | → | R-ant | −1.50 | 0.0007 |
| Semantics | | | | |
| L-ant | → | L-post | 1.08 | 0.0012 |
| L-ant | → | R-ant | 0.97 | <0.0001 |
| L-ant | → | R-post | 1.27 | 0.0026 |
| L-post | → | R-ant | 1.26 | <0.0001 |
| R-ant | → | L-ant | −1.49 | 0.0031 |
| R-ant | → | L-post | −2.45 | 0.0001 |
| R-ant | → | R-post | −3.24 | <0.0001 |
| R-post | → | R-ant | −0.22 | <0.0001 |
| Social cognition | | | | |
| L-ant | → | R-ant | 0.30 | 0.0078 |
| L-post | → | L-ant | 1.23 | <0.0001 |
| L-post | → | R-ant | −0.57 | 0.0013 |
| R-post | → | L-ant | 1.76 | <0.0001 |
| R-post | → | L-post | 1.43 | <0.0001 |
| R-post | → | R-ant | 2.13 | <0.0001 |

*Note:* Significant modulatory parameters ('B-matrix', α ≤0.01) of connectivity between subregions. Based on a random effects permutation test for the null hypothesis 'no parameter difference between tasks'. The strength is given as posterior expectation from the optimum Bayesian parameter average model.

## Discussion

The inferior parietal lobe is a foremost convergence zone of diverse mental capacities, several of which are potentially most developed in the human species. However, it remains open to question how some of the most basic and some of the most advanced cognitive processes converge and diverge in the IPL to realize human communication and interaction. To address this unresolved issue from different perspectives, we carried out a multi-method investigation that makes the functional specialization within the IPL apparent across attention, semantics, and social cognition tasks. By drawing insight from predictive, causal, and functional coupling analyses, we revealed a functional triple dissociation between the probed cognitive domains in the IPL at subregion-resolution. Based on neural activity estimates for the three cognitive domains, two distinctable subregions, one anterior and one posterior, were identified per IPL by a functional parcellation analysis. These subregions differed in their functional specialization across hemispheres and cognitive functions. The right anterior IPL subregion showed the strongest predictive relevance for attentional reorienting while the left anterior IPL subregion was strongest associated with semantic processing. In contrast, both left and right posterior IPL were robustly engaged when subjects adopted the mental perspectives of others. Functional specialization within the larger IPL regions also differed with respect to whole brain connectivity profiles for the different subregions. In general, we observed increased connectivity with diverse large-scale networks with increasing cognitive complexity across tasks. Global connectivity profiles for the anterior IPL subregions showed strong hemispheric lateralization and domain specificity. Functional connectivity profiles for the left and right posterior subregions were similar within each task and showed a higher degree of across-network coupling. Complementary effective connectivity profiles between specialized IPL subregions also differed across domains, supporting the notion of stronger coupling with increasing cognitive complexity. The degree of functional specialization of the IPL has been contemplated before (*Seghier, 2013*). Yet, only in recent years, direct comparison of task experiments from different cognitive neuroscience domains, conventionally studied in isolation, has enjoyed increasing attention (*Bzdok et al., 2013*; *Bzdok et al., 2016*; *Igelström et al., 2016*). Our present study invigorates these beginning endeavors to transcend cognitive fields that are typically studied independently by separate research communities. We strive toward such cross-pollination across disparate literature streams.

To pursue this goal, we show that domain-specific functional signatures became apparent within the IPL when interrogating neural activity responses to tasks, derived predictive principles, as well as induced modulations of directed connectivity among IPL subregions and their functional coupling profiles with distributed brain regions. It is a key advantage that our findings were based on carefully controlled experimental paradigms that were administered to the same subject sample. Consequently, our study offers critical new elements of synthesis that pave the way for a more holistic perspective of IPL specialization.

First, the right IPL showed strong engagement in attentional processes in our study. Previous work has demonstrated a causal relation between tissue damage to the right IPL and left spatial neglect (*Thiebaut de Schotten et al., 2014*). Such neurological patients routinely fail to process cues in their contralesional visual field. Additionally, transient virtual lesions of the right, but not left IPL in the intact human brain were reported to cause performance declines during attentional reorienting (*Rushworth et al., 2001*). Our findings corroborate and detail this previous research by carefully locating attentional processing and accompanying functional coupling shifts preferentially in the anterior subregion of the right IPL, compared to semantics and social cognition. Moreover, neural activity in the right anterior IPL subregion contained information that enabled successfully discriminating attentional reorienting from semantics and social cognition in our predictive modeling analyses. The new insights on right-hemispheric lateralization of attentional reorienting in the IPL were further refined by unique and simple effective connectivity modulations between IPL subregions. Additionally, our analyses of global functional coupling patterns revealed that the right anterior IPL subregion stands out by its connections to bilateral fronto-parietal and visual-sensory networks. Together, these findings speak to a specialized neuronal infrastructure harbored within the right anterior IPL, which is especially tuned to processing demands of attentional reorienting.

Second, our results isolate strong engagement of the anterior subregion of the left IPL for semantic processing. Deficits in various facets of semantic processes, such as in Wernicke's aphasia (*Corbetta et al., 2015*), have received evidence to be closely linked to left temporo-parietal damage

(*Corbetta et al., 2015*; *Dronkers et al., 2004*; *Fridriksson et al., 2016*; *Fridriksson et al., 2010*; *Mirman and Graziano, 2012*; *Schwartz et al., 2011*) among other regions of the language network. Consistently, transient virtual lesions of the left IPL in healthy subjects were reported to disrupt semantic task performance during language comprehension (*Hartwigsen et al., 2017*; *Sliwinska et al., 2015*). Our findings from mapping neural responses during experimental tasks and identifying most predictive subregions reliably situated semantic processing in the left anterior IPL subregion. In addition, our effective connectivity analyses revealed a motif of rich modulatory influences among IPL subregions during semantic processing compared to our other two tasks. Specifically, we found increased task-induced facilitatory coupling from the left to the right IPL, with the left anterior IPL increasing its facilitatory influence on all other subregions during semantic processing. The strong facilitatory modulation exerted from the left to the right IPL was complemented by increased inhibition in the influence of the right on the left IPL and between right-hemispheric IPL subregions, potentially reflecting cognitive control processes (e.g. inhibition of alternative responses). This constellation of task-induced connectivity modulations is consistent with an interpretation of fine-grained interactions between left and right IPL subregions, rather than an exclusive role of the dominant left hemisphere in language processing (see also *Binder et al., 2009*; *Hartwigsen, 2018*; *Hartwigsen et al., 2020*).

Third, adding to the right and left anterior IPL's preferential involvement in attentional and semantic processing, respectively, our perspective taking task highlighted the posterior subregion of the IPL in both hemispheres as key regions for social cognition. This last piece of the functional triple dissociation in the IPL received support from our multivariate predictive algorithm approach, and was further annotated by our effective connectivity analyses. When subjects were concerned with inferring others' mental states, both posterior IPL subregions increased their facilitatory influence on the bilateral anterior IPL subregions. The overall task-specific connectivity configuration derived for our social cognition task was predominantly facilitatory, pointing toward tight intra- and interhemispheric interactions during mental perspective taking. A posterior-to-anterior transition of the neurocognitive processes subserved by the IPL has been proposed in previous research (*Bzdok et al., 2016*; *Gordon et al., 2020*). These authors identified the posterior IPL as relatively higher associative and more domain-general processing hub, compared to the anterior IPL subregions. More broadly, the observed bilateral IPL engagement in widely employed social cognition experiments carefully reconciles previous virtual lesion evidence on the functional relevance of the overall right (*Krall et al., 2016*) or left (*Samson et al., 2004*) IPL region for social cognitive functions.

Our compilation of findings provides a multi-perspective answer to the functional subspecialization of the IPL that spans lower and higher cognitive processes. Previous studies have hinted at or considered the possibility of several specialized subregions in the IPL (*Bzdok et al., 2013*; *Bzdok et al., 2016*; *Kernbach et al., 2018*). While some authors have advocated a monolithic functional role of the IPL, others advertised the possibility of functionally distinct neuronal populations in the IPL, which may be challenging to disentangle using conventional contrast analysis in neuroimaging. To the best of our knowledge, we are the first to conduct a within-subject study across such a diversity of cognitive domains. Thereby, we demonstrate that cognitive processing in the IPL can probably not be accounted for by a single account of functional specialization.

Based on our examination of three functional domains with different experimental tasks, two robust clusters of neural responses within the IPL were identified, each of which probably overlaps with several cytoarchitectonic areas. The observed functional differentiation along the anterior and posterior IPL region is consistent with previous work in human volunteers, which featured such functional gradients based on resting-state connectivity (*Mars et al., 2011*), meta-analytic connectivity modeling (*Bzdok et al., 2016*), and probabilistic fiber tracking (*Caspers et al., 2011*). More anterior portions of the IPL in the left hemisphere were reported to be more specifically associated with lower-level neural processing facets of language and social cognition. Instead, more posterior portions of the IPL were more specifically associated with advanced neural processing facets of both functional domains (*Bzdok et al., 2016*). This observation converges with the present constellation of findings. The relatively automatic lexical decision task in the present study was specifically associated with our left anterior IPL cluster, while the more complex perspective taking task was specifically associated with both posterior IPL subregions. Consistently, our functional connectivity analysis demonstrated relatively stronger across-network coupling with several cortical networks for our posterior IPL cluster.

A functional dissociation in the specialization of anterior and posterior IPL regions is further consistent with findings in non-human primates (*Rozzi et al., 2008*). Anterior parts of the monkey IPL are functionally specialized in somatosensory processes, whereas posterior parts preferentially process information based on visual input (*Kravitz et al., 2011*). The posterior part is thought to be particularly involved in navigation, that is, processing one's position in space (*Crowe et al., 2004*; *Crowe et al., 2005*). This region is further associated with processing visual information in an external, object-centered manner (*Crowe et al., 2008*) compared to an egocentric processing system which is used to control body movements (*Chafee et al., 2007*; *Snyder et al., 1998*). These findings argue for increased processing complexity from anterior to posterior regions in the macaque IPL. It is, however, important to bear in mind that the monkey-human homologue of IPL anatomy remains insufficiently understood (*Mars et al., 2013*; *Seghier, 2013*).

A high degree of abstraction from sensory information in the macaque IPL is further substantiated by the existence of intention- and goal-specific neurons rather than pure movement-coding neurons in the macaque IPL (*Fogassi et al., 2005*; *Rozzi et al., 2008*). Notably, the IPL expanded in the primate lineage (*Orban et al., 2004*), while the existence of a homologue in non-human primate remains uncertain (*Mars et al., 2011*; *Seghier, 2013*). The expansion of the IPL may relate to cognitive capacities which are unique to humans, including the ability for speech and language processing as well as complex problem solving. We confirm and detail the previously reported gradient in the complexity of processing in the human IPL, which likely reflects fine-tuned functional differentiation for higher cognitive operations in humans.

Distinct functional specialization for human-defining cognitive operations in the IPL may explain differences in the connectivity profiles of the larger IPL regions between humans and macaques (e.g. *Margulies et al., 2016*; *Oligschläger et al., 2019*; *Xu et al., 2020*). Uncertain homologies between humans and non-human primates are accompanied by complex structure-function relationships in the human IPL. As an overarching tendency, functional organization reproduces structural organization in unimodal areas in the human brain. In contrast, functional organization is significantly less determined by structure in heteromodal brain regions (*Vázquez-Rodríguez et al., 2019*). Such laxer coupling of structure-function relations is observed in heteromodal association areas, including the IPL (*Margulies et al., 2016*). In general, functional specialization for higher cognitive operations may gradually decouple from the underlying structure (*Vázquez-Rodríguez et al., 2019*). This notion is in line with the present results from our functional parcellation which does not simply follow cytoarchitectonic boundaries.

Our complementary set of findings indicate that task-specific effective connectivity modulations within and from different IPL subregions uniquely characterize each of the probed cognitive domains. For relatively low-level cognitive processes, here exemplified by attentional reorienting, task-specific coupling adjustment implicated a reduced set of coupling partners. In contrast, for more complex tasks, as exemplified by lexical decision and perspective taking, we found evidence for more elaborate coupling motifs, including dense intra- and inter-hemispheric facilitation and inhibition. Notably, there is a scarcity of existing studies directly devoted to investigating the left and right IPL in several different psychological tasks. The few existing studies committed to the effective connectivity of the IPL mainly focused on task-related connectivity changes between an IPL subregion and other areas of a specialized network for a single cognitive domain, typically in one hemisphere only (*Fukuda et al., 2019*; *Hartwigsen et al., 2017*). As such, our results usher toward a broader perspective. We complement the identification of intra- and inter-hemispheric IPL coupling patterns with strong domain-specific coupling profiles from IPL subregions to disparate cortical partners from large-scale brain networks.

Task-induced shifts in functional connectivity revealed key distinctions in interactions with major brain networks. Functional coupling profiles varied significantly between the three experimental tasks. Across domains, coupling partners were recruited from various large-scale brain networks. Our observation reinforces the notion of the IPL interfacing multiple neural systems and different levels of neurocognitive abstraction (*Bzdok et al., 2013*; *Seghier, 2013*).

Moreover, we show that the task-specific coupling profiles between IPL subregions and a variety of cortical partners are anchored in the functional compartments uncovered in the IPL. The wider functional connectivity profiles for the anterior and posterior subregions further elucidate their features of underlying neural processing. As a tendency, functional connectivity profiles for the left and right posterior subregions were similar within each task. Yet, global connectivity profiles varied

between the left and right anterior subregions of the IPL. The hemisphere-specific functional coupling for the anterior IPL subregions is broadly consistent with the common reports of right-hemispheric specialization of attentional reorienting processes (*Rushworth et al., 2001*; *Schuwerk et al., 2017*) and left-hemispheric specialization for semantic processes (*Binder et al., 2009*; *Braga et al., 2019*; *Hartwigsen et al., 2016*). The rich connectivity profiles of the left and right posterior IPL subregions support the notion of bilateral IPL relevance for social cognitive processes, consolidating previous findings (*Bzdok et al., 2013*; *Bzdok et al., 2016*).

Specifically, during attentional processing, the dedicated coupling partners of the right anterior IPL subregion included regions of the visual cortex and the dorsal attention network. During semantic processing, the left anterior subregion enhanced coupling with regions of the default mode and ventral attention network. Ultimately, during social cognition processing, the left and right posterior IPL subregions preferably coupled with the default mode network. The observed task-specific functional coupling of the posterior subregions with major brain networks was largely symmetric across both hemispheres.

To summarize our integrated experimental and computational study, we provide evidence for a functional triple dissociation in the human IPL, with task-specific functional specialization in distinct IPL subregions. This emerging view is supported by information carried in precise multivariate predictive signatures. Zooming into task-specific coupling motifs among IPL subregions, our effective connectivity analyses revealed that attentional reorienting mediated simpler coupling modulations, while semantic and social cognition were realized by more complex inter-hemispheric influences. Delineating task-evoked shifts in functional coupling patterns, in turn, uncovered that the posterior IPL subregions were linked to bilateral, symmetric recruitment of distributed cortical coupling partners, reminiscent of the default mode network. Conversely, the anterior IPL subregions were engaged in flexible and hemisphere-specific coupling patterns with brain-wide cortical partners. Together, our results shed new light on how currently under-appreciated activity and connectivity profiles within the left and right IPL support some of the most distinctive mental capacities in humans.

## Materials and methods

### Subject sample

Twenty-two healthy, native German speakers (11 female, mean age 27.9 ± 3.28 years) participated in this neuroimaging investigation. All subjects had normal or corrected-to-normal vision and no contraindications against magnetic resonance imaging (MRI). Subjects were recruited from the inhouse database at the Max Planck Institute for Human Cognitive and Brain Sciences. Written informed consent was obtained from all subjects before the experiment. All subjects were right-handed (laterality index ≥80% *Oldfield, 1971*). The study was performed according to the guidelines of the Declaration of Helsinki and approved by the Ethics Committee of the Medical Faculty of the University of Leipzig, Germany.

### Experimental design

The functional MRI (fMRI) investigation (see *Figure 1—figure supplement 1A*) consisted of three sessions that were performed on separate days, scheduled at least 7 days apart from each other. Each fMRI session was divided into four runs. In all runs, each of the three tasks (i.e. attentional reorienting, lexical decisions, and perspective taking, see below for details) was administered consecutively in a task block. Within task blocks, trials were presented in an event-related fashion. Each run consisted of 40 attentional reorienting trials (eight *invalid*, 30 *valid*, two *catch* trials), 40 lexical decision trials (20 *word* and 20 *pseudoword* trials), and six perspective taking trials (three *false belief* and three *true belief* trials). Each stimulus for the lexical decision and social cognition task was only shown once per subject, while the attentional reorienting task relied on simple geometric cues that were presented repeatedly. At the beginning of each task block, instructions were presented. Task order was pseudo-randomized. Trial order and timing were defined by a genetic algorithm for optimizing detection power, using an adapted version of Neurodesign v0.2.1 (*Durnez et al., 2018*). All experimental tasks were administered with Presentation Software (v20.1; Neurobehavioral Systems, Berkeley, CA). Prior to the first fMRI session, subjects underwent training for all three tasks outside

the MRI scanner. Training stimuli for the semantic and the social cognition task were not included in the main experiment.

## Tasks

For each probed cognitive domain, we elected a well-established task. Task presentation and response selection were carefully matched across tasks, including visual stimuli and binary choices for all tasks. All three tasks conformed to the principle of using a single target condition and a single control condition to isolate the neural activity of the particular cognitive function. In all three tasks, the subjects responded to a binary choice after each trial via single button press on a two-finger response box with their right hand. Button assignments for the semantic and social cognition tasks were initially pseudo-randomized across subjects, and kept identical across sessions per subject. The button box was fixated on the subject's right thigh to assure a comfortable, natural, and stable lying position in the MR scanner. Subjects were instructed to respond as fast and accurately as possible. Example trials for each of the different tasks and conditions are available in *Figure 1—figure supplement 1*.

### Attention

We used a Posner-like attentional reorienting task, as described previously (*Rushworth et al., 2001*). Each trial started with the presentation of two rectangular, empty boxes (size 2.6° of visual angle, center distance 8.3°, positioned horizontally) and a fixation cross at the center of the screen (*Figure 1—figure supplement 1B*). After an average duration of 3.2 s (*SD* = 1.41 s, minimum inter-stimulus interval = 2 s), the fixation cross was replaced with an arrow pointing either to the right or the left for 250 ms or 350 ms (the 'cue'). Subsequently, a target asterisk was presented in one of the rectangles. In 75% of the trials, the target location was congruent with the direction of the arrow (*valid* condition). In 20% of the trials, the target was presented on the (unexpected) opposite side (*invalid* condition). In 5% of the trials, no asterisk was shown (*catch* condition) and no response was probed to ensure constant attention. Subjects were asked to indicate the location of the target with a button press. Note that the *invalid* condition was intended to recruit attentional reorienting processes and the *valid* condition served as the matched control condition.

### Semantics

We administered a lexical decision task that contrasts words and pseudowords as a prototypical example of semantic processing (*Binder et al., 2003*). 240 concrete German nouns were selected from the SUBTLEX-DE database (*Brysbaert et al., 2011*). Inclusion criteria were (i) two syllables; (ii) frequency per million > 1; (iii) concreteness ratings < 4; and (iv) arousal ratings < 6, according to the LANG rating (note that smaller values correspond to a higher concreteness; *Kanske and Kotz, 2011*). All word stimuli denoted countable entities and were non-ambiguous. A pseudoword was created for each word with Wuggy v0.2.2b2 (*Keuleers and Brysbaert, 2010*) to assure phonotactic validity without semantic content. Pseudowords matched their word counterparts in length, syllable structure, and transition frequencies between subsyllabic elements. Each trial started with the presentation of a fixation cross for at least 2 s (*M* = 3.8 s, *SD* = 1.7). Thereafter, a *word* (target condition) or *pseudoword* (control condition) was shown for 1 s. Subjects were instructed to indicate whether the stimulus represented a word or pseudoword via button press (*Figure 1—figure supplement 1C*).

### Social cognition

For the perspective taking task, we employed an adapted version of the Sally-Anne paradigm, which is known to prompt reasoning about the mental state of others (*Rothmayr et al., 2011*). In total, 72 three-picture comics were presented, with 2 s presentation time per stimulus. Between trials, a fixation cross was shown for at least 2 s (*M* = 3.8 s, *SD* = 1.7 s). In all trials, character A puts an object into a container-like object (picture 1). Then character B passes the object on to another container. Character A either observes this action (*true belief*) or not (*false belief*) (picture 2) and then searches in either the correct or in the wrong location (picture 3). Subjects were instructed to indicate whether the search location was congruent or incongruent with character A's knowledge about the object position via button press (*Figure 1—figure supplement 1D*). A correct response during a *false belief*

trial (target condition) required the subject to infer the mental state of character A from the comic narrative. In contrast, for the control condition (*true belief*) a correct response did not depend on perspective taking and could be accomplished by relying on the physical reality shown to the subjects.

## Functional magnetic resonance imaging

fMRI data acquisition was performed on a 3 Tesla Siemens Prisma system (Siemens, Erlangen, Germany). A whole brain gradient echo planar (GE-EPI) T2* sensitive sequence (3 × 3×3.2 mm, 0.32 mm gap, TR 0.5 s, 36 slices, TE 24 ms, flip angle 45°) with multiband acceleration was used (*Feinberg et al., 2010*). Additionally, a high-resolution (1 × 1×1 mm voxel size) structural MR image (T1w) was acquired for each subject using a standard three-dimensional MPRAGE sequence.

## Preprocessing

The raw fMRI data was despiked with 3dDespike from the AFNI toolbox through Nipype v1.5.0 (*Gorgolewski et al., 2011*). Subsequently, the preprocessing was implemented in fMRIPrep v1.4.1 (*Esteban et al., 2019*), a Nipype-based tool. The individual T1 image was intensity corrected using N4BiasFieldCorrection v2.1.0 (*Tustison et al., 2010*) and skull-stripped using antsBrainExtraction.sh v2.1.0 with the OASIS (*Marcus et al., 2007*) template. Brain surfaces were reconstructed using recon-all from FreeSurfer v6.0.1 (*Dale et al., 1999*). A brain mask was refined to reconcile ANTs-derived and FreeSurfer-derived segmentations of the cortical gray matter in Mindboggle (*Klein et al., 2017*). Spatial normalization to the ICBM 152 nonlinear asymmetrical template version 2009 c (*Fonov et al., 2009*) was performed through nonlinear registration with the antsRegistration tool 2.1.0 (*Avants et al., 2008*). Brain tissue segmentation of cerebrospinal fluid, white matter, and gray matter was performed on the brain-extracted T1 with the fast tool (FSL v5.0.9 *Zhang et al., 2001*).

Functional data was slice-time corrected with 3dTshift from AFNI v16.2.07 (*Cox, 1996*) and motion corrected using mcflirt (FSL v5.0.9 *Jenkinson et al., 2002*). Distortion correction was performed with the TOPUP technique (*Andersson et al., 2003*) using 3dQwarp from the AFNI toolbox. This was followed by co-registration to the T1 using boundary-based registration (bbregister from FSL v6.0.1 *Greve and Fischl, 2009*) with nine degrees of freedom. Motion correction transformations, field distortion correcting warp, functional-to-anatomical transformation and T1-to-MNI warp were concatenated and applied in a single step using antsApplyTransforms (ANTs v2.1.0) with Lanczos interpolation.

To account for motion induced artefacts, physiological noise regressors were extracted with the anatomical version of CompCor (*Behzadi et al., 2007*) (aCompCor). Six components were calculated within the intersection of the subcortical mask and the union of corticospinal fluid and white matter masks. Frame-wise displacement (*Power et al., 2014*) was calculated using the implementation of Nipype.

Statistical Parametric Mapping 12 (SPM 12, Wellcome Department of Imaging Neuroscience, London, UK) was used to spatially smooth the functional data with an 8 mm full-width half-maximum Gaussian kernel.

## Specification of the general linear models

At the single-subject level, two design matrices were specified and general linear models (GLMs) were computed using SPM 12 to estimate task-related neural activity. $GLM_{cond}$ was designed with one regressor per condition in accordance with the standard mass-univariate analysis. The second set of GLMs ($GLM_{trial}$) was purpose-designed for multivariate and task-related functional connectivity analyses with one regressor per trial to gain access to trial-wise neural activity estimates (*Abdulrahman and Henson, 2016*). For $GLM_{cond}$, seven trial regressors were defined (*valid*, *invalid*, *catch*, *word*, *pseudoword*, *false belief*, *true belief*, duration 0 s). Additionally, three rest regressors for task-wise rest periods were included (duration 16 s each). Incorrect responses were modeled separately per task. One regressor per session was added to account for between session variability. To remove high-motion timepoints, one volume masking regressor was added for each volume with a frame-wise displacement value above 0.9 (*Power et al., 2012*), yielding an average of 10.04 masked volumes per subject. Six motion and six aCompCor (*Behzadi et al., 2007*) regressors were included

to account for motion-induced artefacts. The design matrix for GLM$_{trial}$ followed a similar general logic, but included one regressor per trial, leading to trial wise-beta estimates instead of condition-wise beta estimates. Run, motion, and aCompCor regressors were added as above. GLM$_{cond}$ was estimated on smoothed, GLM$_{trial}$ on unsmoothed data. A high-pass filter of 128 s was applied and serial correlations were accounted for with the FAST method (*Corbin et al., 2018*). Global normalization was not performed and the canonical hemodynamic response function was used without derivatives.

## Classical statistical analysis

For each task, two contrasts were defined at the single-subject level, including the condition of interest > rest$_{task}$ (*invalid* > rest$_{att}$, *word* > rest$_{sem}$, and *false belief* > rest$_{soc}$) and control condition > rest$_{task}$ (*valid* > rest$_{att}$, *pseudoword* > rest$_{sem}$, *true belief* > rest$_{soc}$). Task-wise contrasts (*invalid* > *valid*, *word* > *pseudoword*, *false belief* > *true belief*) were computed at the group level.

Brain results were rendered by means of Paraview v5.7.0 (*Ahrens et al., 2005*), circlelize v0.4.8 (*Gu et al., 2014*), and BrainNet Viewer v1.7 (*Xia et al., 2013*). To this end, results were transformed from MNI space (ICBM 152 linear *Mazziotta et al., 2001*) to the surface-based FreeSurfer fsaverage (*Fischl et al., 1999*) coordinate system via the nonlinear mapping Registration Fusion approach v0.6.5 (*Wu et al., 2018*).

## Subregion identification in the IPL

The topographical outline for the left and right IPL ROI was guided by an established and freely available histological atlas. Seven cytoarchitectonic maps cover the human IPL in each hemisphere according to the widely used JuBrain probabilistic cytoarchitectonic atlas (*Amunts and Zilles, 2001*). These anatomical definitions of microstructurally defined areas known to exist in the IPL comprised one rostral (PGp) and one caudal (PGa) region in the angular gyrus, and five regions of the supra-marginal gyrus (PFm, PF, PFop, PGcm, PFt), as provided by the SPM 12 Anatomy Toolbox v2.2b (*Eickhoff et al., 2005*). The outer boundary of the conglomerate of these cytoarchitectonic areas was taken as contours of our ROI in the IPL, separately in the left and right hemisphere. Both of our ensuing ROI definitions occupied similar cortical volume: 1102 voxels in the left and 1123 voxels in the right (*Figure 2A*) hemisphere.

We used tools from machine learning to explore coherent solutions to IPL segregation in the context of our experimental tasks. A *k*-means clustering (*Hartigan and Wong, 1979*) was applied at the voxel level with 1000 random centroid initializations (*Thirion et al., 2014*) based on the estimates from the 'target condition > rest' contrast of the GLM$_{cond}$ in the IPL ROIs. The unsupervised clustering algorithm was applied on pooled information across subjects and separately for each hemisphere. To explore an optimal number of clusters, hemisphere by hemisphere, our choice was anchored in the majority vote taken across 25 distinct cluster validity metrics that were computed for candidate solutions with two to seven clusters using NBClust v3.0 (*Charrad et al., 2014*). Each cluster quality metric provided a ranking of candidate cluster numbers based on a different notion of goodness-of-fit of the candidate solutions. The majority vote of these different ranking criteria provided a principled, data-driven rationale for the final cluster number that we endorsed in our study, in each hemisphere. To ensure robust voxel-to-cluster assignments, all solutions from the 1000 algorithms initializations that resulted in the winning cluster number were merged into one final solution. Specifically, the final assignments of voxel-to-cluster responsibilities were based on agreement across the random algorithm initializations. Voxels with varying cluster allocations across the initializations were excluded from the final cluster solution (cf. Results section).

## Task-predictive information of the IPL subregions

After establishing functionally defined subregions in our IPL ROIs, we directed attention to the predictive information content available at subregion granularity (*Bzdok, 2017*; *Bzdok et al., 2017*; *Bzdok and Ioannidis, 2019*). For this purpose, we carried out a one-versus-rest scheme in combination with a sigmoid-loss (logistic) linear predictive algorithm to automatically detect the task membership of single experimental trials directly from subregion-wise averaged neural activity signals. Model parameter estimation and model evaluation of predictive success were performed conjointly for the three target conditions (*invalid*, *word*, *false belief*). This modeling strategy naturally

yielded a set of predictive weights for all subregions for each task. Appropriately balanced numbers of trials were ensured for all conditions by sub-sampling to the minimum number of trials per subject and fMRI run. We exclusively considered task trials and brain scans with correct responses. Signal deconfounding removed variation that could be explained by session number, session time, subject identify, or run time. A run-wise variable standardization was applied, before model estimation, by de-meaning to zero and unit-variance scaling to one. The generalization performance of the prediction accuracy was estimated via leave-one-subject-out cross-validation. In total, 22 model instances were built from the brain data so that each subject's data were held out once from the training process ('training') and exclusively used to assess the model's generalization properties ('testing'). The parameters of the predictive algorithm were averaged across all cross-validation folds to guard against noise and obtain a single aggregate predictive model solution for inspection and visualization (*Kernbach et al., 2018*). The parameters of the predictive algorithm were averaged across all cross-validation folds to guard against noise and obtain a single final predictive model solution for inspection and visualization (*Kernbach et al., 2018*). The pattern-classification pipelines were realized in the Python data science ecosystem, using especially nilearn v0.6.2 (*Abraham et al., 2014*) and scikit-learn v0.21.2 (*Pedregosa et al., 2011*). Overall, the set of multivariate predictive analyses thus aimed at revealing task-distinctive information available at subregion-level signals in our IPL ROIs.

## Task-evoked functional connectivity shifts

We then quantitatively characterized the task-induced changes in distributed functional coupling profiles as anchored in each of the IPL subregions. Analogous to the multivariate pattern recognition approach (cf. last paragraph), this cortex-wide analysis was performed across the three distinct task contexts. We drew on the commonly used Schaefer-Yeo atlas v0.15.3 (*Schaefer et al., 2018*) with 400 parcels to parse the distributed neural activity estimates ($GLM_{trial}$) of the three target conditions (*invalid*, *word*, *false belief*). Consistent with the other analysis approaches, we discarded any trials with incorrect subject responses. Neural activity estimates were summarized by averaging across all voxels belonging to a given IPL subregion. For each experimental target condition, Pearson's correlation coefficients were computed between a given IPL subregion and each of the Schaefer-Yeo parcels. To rigorously assess whether connectivity links between a specific IPL subregion and cortical parcel were reliably weaker or stronger in one task relative to the other two, a pooled permutation-based baseline was computed across all three task conditions. Task-specific functional coupling shifts were determined by statistical significance testing based on a non-parametric permutation procedure using an empirical null-hypothesis distribution (*Bzdok et al., 2019*; *Bzdok and Yeo, 2017*). The data-derived null model reflected the constellation of neural activity coupling strengths between a given subregion and other cortical regions that would be expected if task A induced similar patterns of brain connectivity, compared to the respective other tasks B and C. Following this fully data-driven pattern-learning tactic, for each of the three tasks, the analysis directly provided brain maps of task-dependent functional connectivity profiles for the IPL subregions.

## Task-specific effective connectivity modulation

After delineating (undirected) functional connectivity of IPL subregions with brain-wide cortical regions, we explicitly examined task-dependent causal interactions within the IPL. To achieve this goal, we analyzed the effective connectivity among the four IPL subregions by means of DCM. Since this approach centered on the directed interaction among the IPL subregions themselves, this analysis selectively included these four compartments. Effective connectivity was estimated with DCM 12.5 (*Friston et al., 2013*) implemented in SPM 12. We defined and deployed subject-wise DCMs with one target node per subregion. Subject-specific node centers per subregion were based on the peak activity across all three tasks. We ensured similar contribution of all tasks by task-wise normalization of neural activity estimates to define the subject-wise DCM nodes. The first eigenvariate from the timeseries of voxels in an 8 mm sphere around this target was extracted (*Zeidman et al., 2019*). We exclusively included data from voxels within the upper quartile of estimated neural activity to ensure functional relevance (*Seghier and Friston, 2013*) and assured that all voxels fell into the respective IPL subregion. A fully connected DCM (full model) estimated all possible connections among the four IPL subregions (including self-connections) and directions of modulatory influences.

The trial onsets of the three target conditions were modeled as direct inputs to the nodes and as modulatory inputs to the inter-regional connections. Neural activity stemming from incorrect trials and control conditions was regressed out (*Zeidman et al., 2019*).

After model estimation, subject-wise full models were optimized for the intrinsic connections (A matrix) and modulatory inputs (B matrix) at the group level using the Bayesian model selection procedure (*Friston and Penny, 2011*). We decided against optimization of the input parameters (C matrix), as we selectively modeled nodes in high-order association cortex. As such, there was no obvious scientific hypothesis regarding the input to the system. The optimized, reduced model represented fixed-effects at the group level (*Rosa et al., 2012*). As a second-level random-effects analysis, we conducted a non-parametric permutation test on subject-wise parameters of the optimal reduced model. This layer of analysis identified those model parameters that varied significantly between tasks. Specifically, for each task, the parameter differences of the remaining two tasks were randomly shuffled 10,000 times to obtain and test against an empirical distribution for the null hypothesis (i.e., no differences between task-dependent effective connectivity). Statistical significance was determined based on the absolute parameter difference higher than 99.9% of the baseline difference, corresponding to $\alpha \leq 0.01$.

## Acknowledgements

This work was supported by the German Research Foundation (BZ2/4-1, BZ2/3-1, and BZ2/2-1 to DB and HA6314/3-1 and HA6314/4-1 to GH), the NVIDIA Corporation (donation of a Titan Xp graphics card to GH), and National Institutes of Health (NIH grant R01AG068563A to DB). DB was further supported by the Healthy Brains Healthy Lives initiative (Canada First Research Excellence fund), by the CIFAR Artificial Intelligence Chairs program (Canada Institute for Advanced Research), and by Google (Research Award). GH was further supported by the Lise Meitner excellence program of the Max Planck Society.

## Additional information

### Funding

| Funder | Grant reference number | Author |
| --- | --- | --- |
| Deutsche Forschungsgemeinschaft | BZ2/4-1 | Danilo Bzdok |
| National Institutes of Health | R01AG068563A | Danilo Bzdok |
| Deutsche Forschungsgemeinschaft | HA 6314/3-1 | Gesa Hartwigsen |
| Deutsche Forschungsgemeinschaft | BZ2/3-1 | Danilo Bzdok |
| Deutsche Forschungsgemeinschaft | BZ2/2-1 | Danilo Bzdok |
| Deutsche Forschungsgemeinschaft | HA 6314/4-1 | Gesa Hartwigsen |
| Max Planck Society | | Gesa Hartwigsen |

The funders had no role in study design, data collection and interpretation, or the decision to submit the work for publication.

### Author contributions

Ole Numssen, Data curation, Formal analysis, Investigation, Visualization, Methodology, Writing - original draft, Writing - review and editing; Danilo Bzdok, Conceptualization, Formal analysis, Funding acquisition, Methodology, Writing - review and editing; Gesa Hartwigsen, Conceptualization, Supervision, Funding acquisition, Investigation, Project administration, Writing - review and editing

## Author ORCIDs

Ole Numssen ⓘ https://orcid.org/0000-0001-7164-2682
Danilo Bzdok ⓘ https://orcid.org/0000-0003-3466-6620
Gesa Hartwigsen ⓘ https://orcid.org/0000-0002-8084-1330

## Ethics

Human subjects: The study was performed according to the guidelines of the Declaration of Helsinki and approved by the Ethics Committee of the Medical Faculty of the University of Leipzig, Germany (282/16-eh). Written informed consent was obtained from all subjects before the experiment.

## Decision letter and Author response

Decision letter https://doi.org/10.7554/eLife.63591.sa1
Author response https://doi.org/10.7554/eLife.63591.sa2

## Additional files

### Supplementary files

- Supplementary file 1. Mass-univariate activation peaks for the three functional domains.
- Supplementary file 2. Intrinsic effective connectivity and task-specific self-connectivity modulation.
- Transparent reporting form

### Data availability

Preprocessed fMRI data and behavioral data are publicly available at the Open Science Framework https://doi.org/10.17605/OSF.IO/9NDHP.

The following dataset was generated:

| Author(s) | Year | Dataset title | Dataset URL | Database and Identifier |
|---|---|---|---|---|
| Numssen O | 2020 | FuncSeg | https://osf.io/9ndhp/ | Open Science Framework, 9NDHP |

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
