## [Decision Letter]

**Acceptance summary:**

This study will be of broad interest to anyone studying posterior parietal cortex. Using both task and connectivity fMRI, the authors use data-driven analyses to argue for a functional parcellation of the human inferior parietal lobule into at least two distinct subregions. The paper provides a strong argument for understanding the broad role of the posterior parietal across tasks, and points at the flexibility of its functional response in supporting those roles.

**Decision letter after peer review:**

Thank you for submitting your article "Hemispheric specialization within the inferior parietal lobe across cognitive domains" for consideration by *eLife*. Your article has been reviewed by two peer reviewers, one of whom is a member of our Board of Reviewing Editors, and the evaluation has been overseen by Chris Baker as the Senior Editor. The reviewers have opted to remain anonymous.

The reviewers have discussed the reviews with one another and the Reviewing Editor has drafted this decision to help you prepare a revised submission.

Summary:

Overall the reviewers felt that the manuscript had a fair amount of promise but raised some issues about the specific tasks used and some details of the analysis. One reviewer in particular felt that the manuscript should be reworked around the functional connectivity results, which would strengthen the manuscript. I tend to agree with this assessment, particularly as concerns the lateralization framing which is not very well explored by these tasks. The following items will need to addressed in order for the paper to move forward.

Essential Revisions:

1) Remove or amend the analysis presented in Figure 3B. If the concern about circularity is unjustified please explain.

2) Please clarify whether the held out data in the analyses came from participants that did not contribute to model creation. If not, please justify why not.

3) Some clarification of the role of the motor commands in the tasks and how they were controlled or may have contributed to the results is necessary.

4) The focus on lateralization in the framing does not seem to be borne out in the data and results that follow. Consider changing the framing to focus on the parcellation.

5) A discussion of the non-human primate literature and its relationship to the findings presented here should be added. This should also include a discussion of why the current results suggest a less differentiated area than suggested by the anatomy or primate work.

Reviewer #1:

This manuscript lays out a series of fMRI investigations and analyses centered on examining the response of the IPL during three different tasks (attention, semantics, social cognition). The analyses are largely data-driven and examine functional response and connectivity, to make the argument for a functional parcellation of the IPL into at least two distinct subregions. The manuscript is well-written and the analyses well described. There are some concerns about the analyses that dampen enthusiasm slightly and a lack of consideration of the associated literature in non-human primates, but these problems seem imminently correctable.

The analyses begin with a data-driven cluster analysis across an anatomically constrained IPL ROI, searching for cluster solutions that efficiently parcellate IPL on the basis of the response of voxels across the three tasks. This analysis is fine, but does constrain the average activity in the identified clusters to differ across the tasks. That makes the univariate activation in Figure 3B a bit circular and hard to interpret. Either the error bars should be removed and a note added that the univariate activity is purely descriptive or the univariate data should be displayed from a slice of the data that did not contribute to the derivation of the clusters. The strongest version of this analysis would hold out entire participants.

The predictive coding analysis is potentially informative but the details were a bit unclear. In the one versus rest analysis the strongest test would be to build the model on the data from n-1 participants and then test it on the trials of the held-out participant. If this was not done, some justification for not doing it would be in order.

Finally, the authors should also consider integrating some of the non-human primate literature as it only strengthens their case. In the human literature the IPL has proved a tough nut to crack, but the single unit physiology has revealed strong differences in the homologous areas of macaque, some of which directly map onto the division argued for here.

Reviewer #2:

In this paper, Numssen and co-workers focus on the functional differences between hemispheres to investigate the "domain-role" of IPL in different type of mental processes. They employ multivariate pattern-learning algorithms to assess the specific involvement of two IPL subregions in three tasks: an attentional task (Attention), a semantic task (Semantics) and a social task (Social cognition). The authors describe how, when involved in different tasks, each right and left IPL subregion recruits a different pattern of connected areas.

The employed tasks are "well established", and the results confirm previous findings. However, the novelty of the paper lies in the fact that the authors use these results as a tool to observe IPL activity when involved in different domains of cognition.

The methodology is sound, well explained in the Materials and methods section, the analyses are appropriate, and the results clear and well explained in the text and in graphic format.

However, a solid experimental design is required to provide strong results. To the reviewer's view, the employed design can provide interesting results about functional connectivity, but not about the functional role of IPL in the investigated functions.

I think the study would be correct and much more interesting if only based on functional connectivity data. Note that rewriting the paper accordingly would lead to a thorough discussion about how anatomical circuits are differently recruited based on different cognitive demands and about the variable role of cortical regions in functional tasks. This issue is neglected in the present Discussion, and this concept is in disagreement with the main results, suggesting (probably beyond the intention of the authors) that different part of the right and left IPL are the areas responsible for the studied functions.

1) the 3 chosen tasks explore functions that are widespread in the brain, and are not specifically aimed at investigating IPL. The results (see. e.g. Figure 1) confirm this idea, but the authors specifically focus on IPL. This seems a rather arbitrary and not justified choice. If they want to explore the lateralization issue, they should consider the whole set of involved areas or use tasks showing all their maximal activation in IPL.

2) The authors aims to study lateralization using an attentional task, considering the violation of a prevision (invalid>valid), a linguistic task, looking for an activation related to word identification (word>pseudoword) and a social task, considering correct prospective taking (false belief>true belief), but they do not consider that in all cases a movement (key press) is required. It is well known that IPL is a key area also for creating motor commands and guiding movements. Accordingly, the lateralization bias observed could be due more to the unbalance between effectors while issuing the motor command, than to a different involvement of IPL regions in the specific tasks functions.

3) Like point 2, the position of keys is also crucial if the authors want to explore lateralization. This is especially important if one consider that IPL plays a major role in spatial attention (e.g. Neglect syndrome). In the Materials and methods, the authors simply say "Button assignments were randomized across subjects and kept identical across sessions", this should be explained in more detail.

4) The authors show to know well the anatomical complexity of IPL, however their results are referred to two large-multiareal-regions. This seems to the reader at odds with all the description related to Figure 2. If they don't find any more subtle distinction within these 2 macro-regions, they should at least discuss this discrepancy.

5) The part about Task-specific network connectivity is indeed very interesting, I would suggest to the authors to focus exclusively on this part. (Note that the results of this part seems to confirm that only the linguistic task is able to show a clear lateralization.

---

## [Author Response]

Essential Revisions:1) Remove or amend the analysis presented in Figure 3B. If the concern about circularity is unjustified please explain.

We agree with the reviewer that the error bars might be misleading to readers of our work. To avoid possible confusion, we have removed the error bars. Additionally, we have moved Figure 3B to the new Figure 3—figure supplement 1. We have also clarified in the figure legend that beta estimates were extracted for visualization purposes from the respective univariate analysis (GLM_cond_) at the center of mass for each IPL subregion.

2) Please clarify whether the held out data in the analyses came from participants that did not contribute to model creation. If not, please justify why not.

We apologize for not being sufficiently clear on this matter. For the linear predictive model (presented in the Results section, subsection “Task-predictive information of the IPL subregions”), we indeed used a one-versus-rest approach to build the model from all but one (n-1) subjects and test it on the held out subject. This building-and-testing sequence was repeated 22 times, that is, one fitted predictive model was applied for each specific subject once to gauge prediction performance. Thereby, data from each subject was used exactly once for the purpose of model evaluation, but never for a model that was fitted with data from this particular subject. The classification accuracy results that we present are the averaged *prediction accuracies* across these 22 model fits (out-of-sample testing) to give an estimate of the model’s performance for unseen or future data. Data visualization in Figure 3 depicts the averaged *model parameters*. This is a very conservative decision as each single model fit is based only on n-1 subjects.

Please see our reply to reviewer 1, comment 2 for specific changes in the manuscript to address this issue.

3) Some clarification of the role of the motor commands in the tasks and how they were controlled or may have contributed to the results is necessary.

The three experimental tasks were chosen to be as similar as possible in terms of the presented stimulus material, while differing in the required cognitive task set (attention, semantics, or social cognition). This setup establishes a consistent response pattern requiring one button press at the end of each trial. During each task one target condition (semantic: *word* condition; attention: *invalid* condition; social cognition: *false belief* condition) was paired with one closely matched control condition (*pseudoword*, *valid*, *true belief*).

To address a potential imbalance in the subjects’ (preparatory) motor responses in the IPL between the three tasks, we carried out an additional univariate SPM analysis. To this end, we specified a new set of GLMs that was based on the original one used for the univariate analysis and functional parcellation (GLM_cond_). Here, we added one regressor that modeled the reaction times for all tasks and conditions. Accordingly, we refer to this newly introduced set of GLMs as GLM_cond+RT_. This reaction time regressor captured neural activity responses linked to potential motor responses, which were not of primary interest in our study. The task-dependent BOLD responses from GLM_cond+RT_ (Figure 2—figure supplement 2) closely resemble the results from the original analysis (Figure 1). We also reiterated the functional parcellation analyses based on these results from the alternative model. A similar parcellation solution with comparable certainty was selected by this approach (Figure 2—figure supplement 3). Consequently, we are confident that our results are valid and the derived interpretations justified.

Please see our reply to reviewer 2, comment 2 for specific results and respective changes in the manuscript. Further details on the button box assignments are given in reply 3 to reviewer 2.

4) The focus on lateralization in the framing does not seem to be borne out in the data and results that follow. Consider changing the framing to focus on the parcellation.

We are thankful for the constructive feedback, which helped us to realize that our previous version of the manuscript put undue emphasis on the lateralization aspect. We have considerably rewritten several parts of our manuscript, including the Abstract, Introduction, and Discussion and also changed the title of our paper to address this concern. Thereby, we have toned down the emphasis in our conclusions on lateralization and instead stress the relevance of our findings for a better understanding of the neural specialization of cognitive functions. We believe that the revised conclusions are better supported by our data. For instance, in the Discussion section, we now summarize our main findings as follows:

“To address this unresolved issue from different perspectives, we carried out a multi-method investigation that makes the functional specialization within the IPL apparent across attention, semantics, and social cognition tasks. […] Complementary effective connectivity profiles between specialized IPL subregions also differed across domains, supporting the notion of stronger coupling with increasing cognitive complexity.”

5) A discussion of the non-human primate literature and its relationship to the findings presented here should be added. This should also include a discussion of why the current results suggest a less differentiated area than suggested by the anatomy or primate work.

We very much welcomed this reviewer suggestion. We have therefore added several new paragraphs to the Discussion section that link the present results to the non-human primate literature on the IPL. We also elaborate on the parcellation results and their potential relationship with previous findings using cytoarchitectonic parcellation approaches in the Discussion.

Motivated by previous work (e.g. Bzdok et al., 2016; Vázquez-Rodríguez et al., 2019), we expected differences in the number of observable subregions in the IPL between known (task-independent) cytoarchitectonic parcellations and a task-dependent functional segregation based on our specific cognitive experiments. For instance, Vázquez-Rodríguez et al., 2019, show that the general structure-function relationship in the human brain is strong in primary sensory and motor regions but significantly less clear-cut for heteromodal areas like the parietal area.

Please also see our reply to reviewer 2, comment 4 for more details on the functional parcellation.

The following passages have been added to the Discussion:

“Based on our examination of three functional domains with different experimental tasks, two robust clusters of neural responses within the IPL were identified, each of which probably overlaps with several cytoarchitectonic areas. […] This notion is in line with the present results from our functional parcellation which does not simply follow cytoarchitectonic boundaries.”

Reviewer #1:This manuscript lays out a series of fMRI investigations and analyses centered on examining the response of the IPL during three different tasks (attention, semantics, social cognition). The analyses are largely data-driven and examine functional response and connectivity, to make the argument for a functional parcellation of the IPL into at least two distinct subregions. The manuscript is well-written and the analyses well described. There are some concerns about the analyses that dampen enthusiasm slightly and a lack of consideration of the associated literature in non-human primates, but these problems seem imminently correctable.The analyses begin with a data-driven cluster analysis across an anatomically constrained IPL ROI, searching for cluster solutions that efficiently parcellate IPL on the basis of the response of voxels across the three tasks. This analysis is fine, but does constrain the average activity in the identified clusters to differ across the tasks. That makes the univariate activation in Figure 3B a bit circular and hard to interpret. Either the error bars should be removed and a note added that the univariate activity is purely descriptive or the univariate data should be displayed from a slice of the data that did not contribute to the derivation of the clusters. The strongest version of this analysis would hold out entire participants.

In response to this comment, we have removed the error bars from the figure and moved the respective panel to the new Figure 3—figure supplement 1 to avoid confusion. Please see our reply to comment 1 of the Essential revisions above for more details.

The predictive coding analysis is potentially informative but the details were a bit unclear. In the one versus rest analysis the strongest test would be to build the model on the data from n-1 participants and then test it on the trials of the held-out participant. If this was not done, some justification for not doing it would be in order.

We apologize for not being clear enough on this important point. For the linear predictive model (described in the Results subsection “Task-predictive information of the IPL subregions”) we indeed used a one-versus-rest approach to build the model from all but one (n-1) subjects and test it on the held out subject. This building-and-testing sequence was repeated 22 times, that is, once for each subject. Thereby, data from each subject was used exactly once for the model validation, but never for a model that was fitted with data from this particular subject. The classification accuracy results that we present are the averaged (out-of-sample) *accuracies* yielded for the 22 model fits to give an estimate of the models performance for unseen data. Visualization of the results in Figure 3 depicts the averaged *model parameters*. This is a very robust estimate of what the model has learned from the data as each single model fit is based only on n-1 subjects. We revised the respective passages as follows:

Results subsection “Task activity responses are predictive for different cognitive domains”: “We evaluated the quality of the predictive model via leave-one-subject-out cross-validation, that is, by testing the fitted model on data from a subject that was not used during model building.”

Materials and methods subsection “Task-predictive information of the IPL subregions”:

“In total, 22 model instances were built from the brain data so that each subject’s data were held out once from the training process (‘training’) and exclusively used to assess the model’s generalization properties (‘testing’). The parameters of the predictive algorithm were averaged across all cross-validation folds to guard against noise and obtain a single aggregate predictive model solution for inspection and visualization (Kernbach et al., 2018).”

Figure 3. Task-specific predictive contributions of IPL subregions: “A leave-one-subject-out cross-validation was implemented to fit the predictive model.”

Finally, the authors should also consider integrating some of the non-human primate literature as it only strengthens their case. In the human literature the IPL has proved a tough nut to crack, but the single unit physiology has revealed strong differences in the homologous areas of macaque, some of which directly map onto the division argued for here.

Thank you for pointing us to the non-human primate literature. We now link our findings to previous work in non-human primates as follows:

Discussion:

“A functional dissociation in the specialization of anterior and posterior IPL regions is further consistent with findings in non-human primates (Rozzi et al., 2008). […] Uncertain homologies between humans and non-human primates are accompanied by complex structure-function relationships in the human IPL.”

Reviewer #2:In this paper, Numssen and co-workers focus on the functional differences between hemispheres to investigate the "domain-role" of IPL in different type of mental processes. They employ multivariate pattern-learning algorithms to assess the specific involvement of two IPL subregions in three tasks: an attentional task (Attention), a semantic task (Semantics) and a social task (Social cognition). The authors describe how, when involved in different tasks, each right and left IPL subregion recruits a different pattern of connected areas.The employed tasks are "well established", and the results confirm previous findings. However, the novelty of the paper lies in the fact that the authors use these results as a tool to observe IPL activity when involved in different domains of cognition.The methodology is sound, well explained in the Materials and methods section, the analyses are appropriate, and the results clear and well explained in the text and in graphic format.However, a solid experimental design is required to provide strong results. To the reviewer's view, the employed design can provide interesting results about functional connectivity, but not about the functional role of IPL in the investigated functions.I think the study would be correct and much more interesting if only based on functional connectivity data. Note that rewriting the paper accordingly would lead to a thorough discussion about how anatomical circuits are differently recruited based on different cognitive demands and about the variable role of cortical regions in functional tasks. This issue is neglected in the present Discussion, and this concept is in disagreement with the main results, suggesting (probably beyond the intention of the authors) that different part of the right and left IPL are the areas responsible for the studied functions.1) The 3 chosen tasks explore functions that are widespread in the brain, and are not specifically aimed at investigating IPL. The results (see. e.g. Figure 1) confirm this idea, but the authors specifically focus on IPL. This seems a rather arbitrary and not justified choice. If they want to explore the lateralization issue, they should consider the whole set of involved areas or use tasks showing all their maximal activation in IPL.

In response to this reviewer comment, we have toned down the conclusions on lateralization and now focus on functional specialization. As stated in the reply to comment 4 of the Essential revisions above, we have rewritten parts of the Abstract, Introduction, and Discussion to address this concern.

The focus of our study on the human IPL was motivated by the suggested key contribution of this heteromodal association area across numerous cognitive domains (e.g. Bzdok et al., 2016; Seghier, 2012). We acknowledge that many, if not all, (cognitive) functions are embedded in large-scale networks in the human brain. Our main interest was to explore task-specific contributions of different IPL subregions and elucidate their interactions with whole-brain networks. We would argue that our focus on three prototypical cognitive domains in the human brain is well suited to address this question. Based on the previous literature, we think that there is ample evidence for a strong IPL contribution to the three selected domains, which is supported by the univariate analyses, the parcellation approach, and the connectivity analyses. We hope that the motivation for the choice of our tasks, approaches, and region-of-interest is more convincing now.

2) The authors aims to study lateralization using an attentional task, considering the violation of a prevision (invalid>valid), a linguistic task, looking for an activation related to word identification (word>pseudoword) and a social task, considering correct prospective taking (false belief>true belief), but they do not consider that in all cases a movement (key press) is required. It is well known that IPL is a key area also for creating motor commands and guiding movements. Accordingly, the lateralization bias observed could be due more to the unbalance between effectors while issuing the motor command, than to a different involvement of IPL regions in the specific tasks functions.

The three experimental tasks were chosen to be as similar as possible in terms of the presented stimulus material, while differing in the required specific cognitive task set (attention, semantics, or social cognition). This setup establishes a consistent response pattern requiring one button press at the end of each trial. During each task one target condition (semantic: *word* condition; attention: *invalid* condition; social cognition: *false belief* condition) was paired with one closely matched control condition (*pseudoword*, *valid*, *true belief*).

To address a potential imbalance in the subjects’ (preparatory) motor responses in the IPL between the three tasks, we carried out an additional univariate SPM analysis. To this end, we specified a new set of GLMs that was based on the original one used for the univariate analysis and functional parcellation (GLM_cond_). Here, we added one regressor that modeled the reaction times for all tasks and conditions. Accordingly, we refer to this newly introduced set of GLMs as GLM_cond+RT_. This reaction time regressor captured neural activity responses linked to potential motor responses, which were not of primary scientific interest in our study. The task-dependent BOLD responses from GLM_cond+RT_ (Figure 2—figure supplement 2) closely resembled the results from the original analysis (Figure 1). We also reiterated the functional parcellation analyses based on these results from the alternative model. A similar parcellation solution with comparable certainty was selected by this approach (Figure 2—figure supplement 3). Consequently, we are confident that our results are valid and the derived interpretations justified. Note that we have uploaded the unthresholded whole-brain statistical maps from both models to neurovault.org (https://neurovault.org/collections/9350/) for the sake of transparency and re-use by other investigators.

Results subsection “Functional parcellation of the IPL in two subregions”: “To investigate the possible influence of general motor responses within the IPL on our results, we estimated a third set of GLMs (GLM_cond+RT_), which was based on the initial GLM_cond_ with one additional regressor that modeled reaction times for all three tasks. […] The cluster similarity across solutions is illustrated in Figure 3—figure supplement 3.”

3) Like point 2, the position of keys is also crucial if the authors want to explore lateralization. This is especially important if one consider that IPL plays a major role in spatial attention (e.g. Neglect syndrome). In the Materials and methods, the authors simply say "Button assignments were randomized across subjects and kept identical across sessions", this should be explained in more detail.

Thank you for pointing this out. When subjects were prepared for MR scanning, the two-finger button response box was fixated at their right thigh to assure a stable and comfortable position. We deliberately chose to keep the response hand constant across subjects to avoid potential differences between the dominant and non-dominant hand as all subjects were right-handed. The button assignments for the semantic task (*word* or *pseudoword*) and the social cognition task (*false belief* or *true belief*) were pseudo-randomized across subjects. For each subject, the button assignments were kept constant across the three experimental sessions.

We updated the task description to provide detailed information on the response button configuration as follows:

Materials and methods subsection “Tasks”:

“In all three tasks, subjects responded to a binary choice after each trial with a single button press on a two-finger response box with their right hand. […] The button box was fixated on the subject’s right thigh to assure a comfortable, natural, and stable lying position in the MR scanner.“

4) The authors show to know well the anatomical complexity of IPL, however their results are referred to two large-multiareal-regions. This seems to the reader at odds with all the description related to Figure 2. If they don't find any more subtle distinction within these 2 macro-regions, they should at least discuss this discrepancy.

Thank you for alerting us to this issue. From a methodological perspective, we would argue that a parcellation strongly depends on the selected method and the input data. The data driven clustering (and cluster number determination) analysis was based on functional activity for the three examined tasks. Therefore, only subregions that significantly differ in their response profile for the domains attention, semantics, and social cognition could be identified. This analysis was carried out with the repeated use of NBClust, a software package that implements 25 metrics to assess the quality of clustering solutions, to prevent possible biases due to the selection of a single cluster metric and its specific criterion. Thereby, we explicitly tested the validity of parcellations of up to seven clusters per IPL to allow for more fine-grained solutions.

We wish to note that a direct structure-to-function mapping is usually not reported in heteromodal association areas such as the human IPL. This contrasts with primary somatomotor areas in which the functional organization reproduces the underlying structural organization more closely (Vázquez-Rodríguez et al., 2019).

We have updated the Discussion of the manuscript to include this argumentation as follows:

“Uncertain homologies between humans and non-human primates are accompanied by complex structure-function relationships in the human IPL. […] This notion is in line with the present results from our functional parcellation which does not simply follow cytoarchitectonic boundaries.”

5) The part about Task-specific network connectivity is indeed very interesting, I would suggest to the authors to focus exclusively on this part. (Note that the results of this part seems to confirm that only the linguistic task is able to show a clear lateralization.

We agree with the reviewer that the task-specific network interactions provide new insight into the functional specialization of different IPL subregions across three key cognitive domains. Please note that the connectivity analysis directly builds on the parcellation results. We think that the different analyses are complementary and contribute to a better understanding of the functional specialization of different IPL subregions. First, a parcellation was performed to distinguish different subregions based on their predictive relevance for each of the three cognitive domains. The functional whole-brain connectivity analyses show that the different domains also differ with respect to their whole brain connectivity patterns. Moreover, these analyses also show a general difference in laterality between anterior and posterior IPL subregions. Finally, the effective connectivity analyses zoom into the functional interactions between IPL subregions and provide further evidence for an increase in connectivity with increasing task complexity. As noted above, we have toned down the laterality argument and rephrased parts of the Discussion to emphasize the differences in functional specialization of different IPL subregions and their connectivity profiles.